# Radar-Spectrogram-Based UAV Classification Using Convolutional Neural Networks

**DOI:** 10.3390/s21010210

**Published:** 2020-12-31

**Authors:** Dongsuk Park, Seungeui Lee, SeongUk Park, Nojun Kwak

**Affiliations:** Department of Intelligence and Information, Seoul National University, 1 Gwanak-ro, Gwanak-gu, Seoul 08826, Korea; ds.park@snu.ac.kr (D.P.); seungeui.lee@snu.ac.kr (S.L.); swpark0703@snu.ac.kr (S.P.)

**Keywords:** CNN, classification, UAV, FMCW radar, STFT, spectrogram, MDS

## Abstract

With the upsurge in the use of Unmanned Aerial Vehicles (UAVs) in various fields, detecting and identifying them in real-time are becoming important topics. However, the identification of UAVs is difficult due to their characteristics such as low altitude, slow speed, and small radar cross-section (LSS). With the existing deterministic approach, the algorithm becomes complex and requires a large number of computations, making it unsuitable for real-time systems. Hence, effective alternatives enabling real-time identification of these new threats are needed. Deep learning-based classification models learn features from data by themselves and have shown outstanding performance in computer vision tasks. In this paper, we propose a deep learning-based classification model that learns the micro-Doppler signatures (MDS) of targets represented on radar spectrogram images. To enable this, first, we recorded five LSS targets (three types of UAVs and two different types of human activities) with a frequency modulated continuous wave (FMCW) radar in various scenarios. Then, we converted signals into spectrograms in the form of images by Short time Fourier transform (STFT). After the data refinement and augmentation, we made our own radar spectrogram dataset. Secondly, we analyzed characteristics of the radar spectrogram dataset with the ResNet-18 model and designed the ResNet-SP model with less computation, higher accuracy and stability based on the ResNet-18 model. The results show that the proposed ResNet-SP has a training time of 242 s and an accuracy of 83.39%, which is superior to the ResNet-18 that takes 640 s for training with an accuracy of 79.88%.

## 1. Introduction

In recent years, the rapid development of Unmanned Aerial Vehicle (UAV) technology has increased the usage of UAVs in various fields such as agriculture, industry, and military. Even though the use of UAVs brings convenience to life, it poses severe threats if abused by enemies or terrorists. There have been reports of attempts to assassinate key figures or to attack oil facilities using UAVs loaded with small bombs. If UAVs are used to attack with biochemical weapons, damages will be more severe. Therefore, real-time early detection and identification of UAVs are essential in real-life scenarios. However, it is difficult to identify UAVs due to their low altitude, slow speed, and small radar cross-section (LSS) characteristics. In the case of the deterministic rule-based model, due to the LSS characteristics of UAVs, the algorithm becomes more complex, which increases the number of computations. Hence, effective alternatives enabling real-time identification of these new threats are required.

Recently, deep learning-based classification models have been used to solve various tasks. These models learn features from a large amount of data themselves and have shown outstanding performance in various classification tasks. In the UAV classification task, deep learning-based classification models using various sensors are being actively studied. Saqib et al. [1] experimented the task of UAV detection using pre-trained models such as ZF-Net [2] and VGG-16 [3] with the bird-vs-UAV dataset, which contains 5 videos with 2727 frames. It resulted the highest mean average precision of 0.66 with VGG-16. Kim et al. [4] performed Fast Fourier Transform (FFT) on the real-time acoustic data and tested the performance by applying the plotted image machine learning (PIL) and the k-nearest neighbors (KNN) methods, reaching accuracy of 83% and 61% in PIL and KNN, respectively. Seo et al. [5] recorded acoustic signals of UAVs and non-UAVs outdoors, obtained a 2D image by applying STFT, and tested the dataset with a self-designed CNN model. The result showed a detection rate of 98.97% and a false alarm rate of 1.28%.

Radar poses some advantages over other sensors. Namely, radar is less affected by weather and low visibility environments than optical sensors. Unlike acoustic sensors, it is not vulnerable to ambient noise. Because of these advantages, various deep learning-based UAV classification tasks such as birds vs. UAVs, UAVs vs. UAVs and UAV characterization have been conducted using radars [6]. Among them, there exists a research field that focuses on classification using deep learning by learning the MDS of a moving target as the main feature. A moving target generates the micro-Doppler effect by partial movements like the pendulum, rotation and vibration along with a constant Doppler shift induced from the main body. This MDS is a unique characteristic of the target and is visually represented well in a spectrogram of the STFT of the radar signal. This spectrogram can be trained with a deep learning image classification model.

Choi et al. [7] suggested a deep-learning model which classifies three types of UAVs (Vario helicopter, DJI Phantom 2, and DJI S1000+) based on the micro-Doppler signatures in the spectrogram and confirmed the feasibility of the application of deep learning-based models in the UAVs classification. Raman et al. [8] proposed a radar spectrogram-based deep learning model that classifies birds and UAVs. They applied the following methods to mitigate the lack of diversity and quantity of UAV radar spectrogram data. First, they added the flying dynamics of UAVs for diversify the dataset. They added more flight dynamics such as radial traversing, pointing out that other previous studies such as [7] only utilized the hovering data of UAVs. Second, they applied the transfer learning [9] commonly used in the optical image classification to solve the lack of radar spectrogram data. Transfer learning is a method that can improve performance by transferring well-trained parameters of a network trained with a large dataset to the network with a small amount of data. To apply this method, the authors transformed the radar signal into an RGB spectrogram of the same color scale as the optical image and trained the dataset with the modified GoogleNet [10]. They showed high performance of over 99%. However, radar signals have very different characteristics from optical images. In particular, data characteristics may be distorted or omitted in the process of transforming the color scale of radar spectrogram data to suit the optical image classification network.

In this study, we generated a radar spectrogram dataset with a variety of UAV flight dynamics and designed a lightweight deep learning classification model that learns the MDS of targets, suitable for the real-time system. To diversify the dataset, radar signals were recorded by diversifying the target type and the movement assuming a real-life scenario. We recorded five LSS targets (3 types of UAVs and 2 different human activities) each with an FMCW radar. UAV targets were selected according to flight type (multicopter, fixed wing, wing flap) and walking and sit-walking were chosen as ground moving targets. UAV signals were recorded by changing altitude, speed, and direction, and human signals were recorded by changing direction and range at a constant walking speed. The signals were converted into spectrogram images through STFT. Then, through the data refinement and augmentation, we generated the radar spectrogram dataset. Then, we analyzed characteristics of radar spectrogram data using the ResNet-18 model [11], which is a popular image classification model. With this model, we analyzed the performance according to the radar spectrogram data type and the model structure. Based on this, we designed a lightweight ResNet-SP model which is more suitable for real-time systems. Additionally, we improved model’s stability by applying anomaly detection and gradient clip methods to reduce learning instability caused by abnormal data. The results show that ResNet-SP has 83.39% of accuracy which is higher than 79.88% from the ResNet-18. Also, the training time is 242 s with our proposed model, which is faster than 640 s of ResNet-18. Furthermore, the ResNet-SP model is more stable through out the training process.

## 2. Micro-Doppler Signature (MDS)

The Doppler effect of a radar is a frequency shift or wavelength change generated from the reflected radar signal when a target moves or changes in a relative distance to an observer. Radar signal interacts with the target in motion and the returned signal changes its characteristics. While the Doppler effect is generated by a bulky motion of the body of a target, its micro-movements from the part of the body can generate such micro-frequency shifts, which is called the micro-Doppler effect [12]. This micro-Doppler signal is created by all subtle movements of a target, such as vibration, rotation, pendulum, etc., unique patterns or characteristics occur depending on the object type or different movements of the same object. Figure 1 shows the micro-Doppler signature of walking represented on the radar spectrogram. On this spectrogram, we can see the MDS shape generated by swinging limbs around the torso signal. The shape of this MDS is represented differently depending on the length of the limb, the swinging period, and the angle. Hence, we can use this MDS shape as the main feature of the classification task.

In UAV, MDS appears differently according to the flight types, and even within the same flight type, it appears differently depending on the number of rotors, blade length, etc. Figure 2 is a radar spectrogram for UAVs of different flight types. The second column shows the spectogram obtained by setting the UAV to the max speed with the fuselage fixed at short-range, and the third column is the spectrogram obtained during free-flight. We can see that each UAV spectrogram appears differently and using this characteristic, we can further perform deep learning-based UAV classification.

## 3. Dataset Generation

UAV flight data is mainly generated by radar-related companies, agencies, or the military for particular purposes. Thus, datasets are not publicly released and it is difficult to find published references. In particular, the dataset recorded by radar sensors is even rarer. Due to characteristics of the research field, many researchers generate their own datasets and carry out research. However, most of datasets do not reflect the diversity of data, such as measuring only at a close range to obtain a clear signal or simulating only limited movements of UAVs. Unlike these datasets, we recorded various types of UAVs and ground moving targets in diverse scenarios. After pre-processing the well-recorded signals, we generated our radar spectrogram dataset.

### 3.1. Measurement

Radar signals are less vulnerable to low visibility and weather conditions than video signals and have fewer restrictions on the line of sight (LOS), which indicates a straight line between the target and the sensor. These radars are divided into two types by the principle of radio wave emission; (1) ‘pulse radar,’ which transmits pulse signals and receives signals reflected from objects and (2) ‘continuous wave (CW) radar’, which continuously transmits and receives signals without a pause. To detect time-varying changes for low radar cross section (RCS) targets, the continuous wave radar is suitable and we decided to use an FMCW radar that continuously emits a frequency modulated signal at regular intervals to obtain time information. Our model is Ancortek’s SDR KIT 980AD2 and the specifications are described in Figure 3. Additionally, to select well-recorded files, we installed a video camera synchronized with the radar and double-checked video files and radar spectrograms.

We recorded five different LSS targets with the FMCW radar. Assuming the enemy approaching the local area, we selected three types of UAVs as aerial moving targets and two different human activities as ground moving targets. The three flight types of UAVs are ‘Metafly’, a wing-flapping drone that mimics wings of a bird and ‘Disco’, a fixed-wing, and ‘Mavic Air 2’, a quad-copter (4 rotors). ‘Walking’ and ‘Sit-walking’ are data of the same person. Figure 4 shows the images of the five targets.

We recorded various movements of targets within the 100 m range. UAVs were recorded while changing altitude, speed, and direction freely, and humans were recorded while changing the distance and the direction at a constant pace. Only two UAVs (Metafly and Disco) were given some restrictions for the proper recording. Metafly was recorded within the 10 m range because of its low signal intensity. Disco was recorded only in the left and right, front and rear, and concentric circular flight at an altitude of 10 m with low-velocity settings because of its high-speed and wide turning radius. Disco is equipped with a single rotor at the rear of the fuselage so that thrust acts only forward and changes direction gradually by changing the Angle of the Attack (AoA) of the aileron at the wing-tips. So it requires a wide turning radius and often be placed outside of the radar’s detection range. Besides, because it moves at high speed, it quickly leaves the radar’s detection range. Table 1 shows the movements for each target and the settings for recording. Figure 5 is sequential video frames of a specific movement for targets.

We operated Metafly and Mavic Air 2 manually, and Disco operated automatically by entering flight plans through the ‘Free Flight Pro’ mobile application. We recorded many times for each target and removed abnormal files such as overly noisy files or files intruded by other objects by cross-checking video files and spectrograms. Basically, we selected 10 well-recorded files for each target and divided the training dataset and the test dataset by a ratio of 8:2. (The exception is for Disco; 25 files were used because the recorded section was too short).

### 3.2. Pre-Processing

In the pre-processing step, the recorded radar signals are transformed into spectrogram images through STFT and completed into the dataset after the data refinement and augmentation. The data refinement is the step for removing the spectrogram section in which the target is not recorded. To do this, we cut the spectrogram into short time intervals and removed cut images with an average intensity below a threshold. To increase the amount of data, we applied three data augmentation methods, keeping the format of the spectrogram: the x-axis represents time, the y-axis represents frequency and the color at each point represents the amplitude of a specific frequency at a specific time.

In the signal processing of STFT, we applied different window sizes (128, 256 and 512) and the window overlap ratios (50%, 70% and 85%) to get spectrograms of different resolutions. In addition, we applied the vertical flip after the data refinement to obtain spectrograms with reversed radial velocity sign.

A spectrogram [13] reveals the instantaneous spectral content of the time-domain signal and the spectral content variations over time. A spectrogram is obtained by the squared magnitude of the STFT of a discrete signal. With the spectrogram, we can visually observe the spectrum of frequency changing over time. But when converting the spectrogram, finite-size sampling in a recorded signal may result in a truncated waveform from the original continuous-time signal, introducing discontinuities into the recorded signal. These discontinuities are represented in the FFT as high-frequency components, even though not present in the original signal. This appears as a blurry form, rather than a clear form on the spectrogram. This is called ‘spectral leakage’ because it looks as if energy is leaking from one frequency to another. In order to mitigate the spectral leakage, window functions are generally applied. The spectrogram resolution is determined by the window size and there is a trade-off between time and frequency resolution [14]. Figure 6 shows the differences in the spectrogram resolution according to window sizes.

If a narrow window size is applied, a fine time resolution can be obtained due to a short time interval, but the frequency resolution is degraded due to the wide frequency bandwidth. Conversely, if wide window size is applied, a fine frequency resolution is obtained due to a wide time interval and a narrow frequency bandwidth, but the time resolution is degraded. The higher the resolution, the more detailed the object’s MDS waveform is represented. We generated spectrogram images with different resolutions by applying three window sizes (128, 256, 512) to the original signal.

Even when the window size is determined, if several different frequencies are included in a window, they may not be distinguishable. One can use a window overlap that applies for redundancy when applying the next window in the STFT process to reduce this effect. The higher the overlap ratio is applied, the higher the resolution, but it requires more computations. Figure 7 shows the differences in the spectrogram resolution of Metafly (wing flapping UAV) according to different window overlap ratios. The higher the overlap ratio in the given window size, the more detailed the MDS signal is. The trajectory of radial velocity by the entire body of the target is also precisely expressed.

In the time-velocity spectrogram, the height represents the target’s radial velocity relative to the radar; the radial velocity component that appears on the upside (positive velocity) from the center represents the target is moving away from the radar, the downward (negative velocity) from the center represents that the target is moving toward the radar and center represents velocity zero. The continuous waveform of the target over time generates a trajectory representing the movement characteristics according to the type of target on the spectrogram. For example, the difference in trajectory due to flight dynamics between fixed-wing aircraft and multiple helicopters is explained below. First, in fixed-wing UAVs, the propeller is fixed in the front or rear, so the thrust works only in one direction. Accordingly, the direction changes gradually by three factors; the inclination of the aileron at the rear of the main wing, the elevator of the horizontal tail wing, and the rudder of the vertical tail wing. In contrast, in a multi-copter, several rotors are distributed over the top of the fuselage. When changing the direction, it uses fuselage-tilting caused by the difference at each rotor rotation rate, so not only a gradual change of the direction but also a drastic change of the direction in all azimuth is possible. These distinctive flight characteristics appear as time-varying trajectories on the spectrogram; in the former case, it is gradual and curved and in the latter case, it appears in a sharp and vertical form. This trajectory will be trained with the target’s characteristics along with the spectrogram shape and the spectrogram with a high overlap ratio will represent the radial velocity change in more detail. We applied three window overlap ratios for each window size. In the STFT process, data augmented nine times by applying three window sizes and three window overlap ratios to one original signal.

We performed data refinement after STFT. UAV signal has low intensity due to its small size and material such as plastic or reinforcement styrofoam. So, as the distance increases, the signal intensity drops sharply or is not detected at all. So there are many unrecorded sections like background clutter in the spectrogram. Figure 8 shows the spectrogram for the background clutter and Mavic Air 2.

In the spectrogram of Mavic Air 2 on the right, the red box is non-recorded sections because of the target’s low signal intensity. When these non-recorded sections are trained with data, it is hard to expect the correct performance of the deep learning model. So we applied the following data refinement process to remove abnormal data. If the target is well captured, the clear spectrogram shape with strong intensity appears around a specific velocity component on the spectrogram and harmonic components are represented parallel around it. Based on this property, we first chopped the image at a time interval, which is the MDS periodicity of the target. Then, we removed chopped images with an average intensity below the threshold and stitched chopped images with an average intensity above the threshold. If the threshold is too high, only high-intensity signals recorded at a short-range would be retained and low-intensity signals at a long-range could be removed even though the MDS shape was represented. Conversely, if the threshold is too low, non-recorded sections of the target cannot be removed. So we determined the threshold by referring to the average intensity values of the background clutter and non-recorded sections of UAV spectrograms. Figure 9 represents the data refinement process for the Mavic Air 2 spectrogram. The spectrogram is cut at the same time interval, and the cut images with average intensity below threshold (red-box) are removed. Images above the threshold (blue boxes) are stitched together to generate a refined spectrogram. MDS periodicity (approx.): Walking (1/2 s), Metafly (1/24 s), Mavic Air 2 (1/92 s), Disco (1/183 s)

The data refinement process was applied to only two UAV targets (Mavic Air 2 and Disco) with many non-recorded sections on the spectrogram. Table 2 shows the change in the spectrogram size before and after the refinement for these two targets. After refinement, the spectrogram size of Mavic Air 2 was reduced by about 25 % and the Disco by about 50%.

In particular, the spectrogram size of the Disco was significantly reduced due to the flight characteristics of fixed-wing UAV. Disco has a single rotor mounted at the rear of the fuselage, so the thrust acts only forward, and the direction changes gradually by the ailerons at the wing-tips. Therefore, it requires a wide turning radius, which often leaves the radar detection range. Besides, its RCS is very low because of the fuselage material which is reinforced styrofoam. In other words, it was difficult to record due to the low RCS, and due to flight characteristics such as high-speed movement and wide turning radius, it was within the radar detection range only for a short time. The data refinement process resulted in increased model stability. Figure 10 is the training loss curve before and after data refinement. In training with unrefined data, accuracy often fell significantly during training and the test accuracy also had a large deviation. You can see this by the number and size of spikes in the training loss curve on the left. Conversely, with refined spectrogram data, the phenomenon of drastic accuracy drop during training was and the variation of test accuracy were reduced. This can be seen in the figure on the right as the size and number of spikes decreased.

After the refinement process, we applied the vertical flip to spectrograms. By using the vertical flip, we got additional spectrograms with reversed radial velocity sign. Totally we could generate 18 different spectrograms from one original radar signal by applying three window sizes, three window overlap ratios and vertical flip. Table 3 shows applied data augmentation methods when generating the training data. The test dataset was generated by applying only one window size (128) and the overlap ratio (70%), without using the data augmentation.

For the training data, after pre-processing, the height of each spectrogram is resized to 128 and then cut into a 128 × 128 spectrogram image by applying a 50% overlap ratio. The test data is cut into a 128 × 128 spectrogram image by applying a 75% overlap ratio after the pre-processing process. To prevent the class imbalance, the number of each class of the training data and the test data was balanced. The number of examples for each class was set to about 2000 in the training data and about 200 in the test data. Table 4 shows the number of examples for each class in our dataset.

## 4. Models

This section introduces the deep learning model and analyzes characteristics of the radar spectrogram dataset using the ResNet-18 model, a popular image classification model. By checking model performances depending on the data type of radar spectrogram and the noise, we confirm the optimal data type and a major feature of the spectrogram dataset. In addition, we check the performance by changing the structure of the model. We design a lightweight and stable ResNet-SP model which is suitable for real-time systems by modifying the model, based on these characteristics.

The rule-based classifier is based on ‘if-then’ rules designed by engineers. This method often complicates the model and lacks scalability, because rules must be specified every time to classify the new data. On the other hand, a CNN-based classifier extracts features from large amounts of data automatically. In addition, the robustness of CNN to shift and distortion [15] resulted in an outstanding performance in image classification tasks; in the 2014 ImageNet Large Scale Visual Recognition Challenge (ILSVRC), GoogLeNet and VGG-Net respectively ranked first and second with top-5 error rates of 6.67% and 7.3% and In the 2105 ILSVRC, ResNet recorded a recognition top-5 error rate of 3.57% which was less than the human recognition error rate.

CNN is a deep-learning model that uses convolutional operation, and is composed of several convolution layers and pooling layers. A model learns the different features of an image using various sizes and numbers of kernels. In the shallow layers, low-level features are learned, which can be basic shapes like lines and edges. In the deep layers, high-level features are learned, which contain more specific information for classifying objects. The model is designed to perform well by learning the various characteristics of the data. CNN can extract high-level features as layers are stacked deeper, but simply stacking layers deeper does not increase the performance. The reason is known to be the gradient vanishing problem [16] that occurs due to the multiplications of gradients in the parameter update stage as the layer gets deeper. As a result, training cannot be progressed and in some cases, the performance even degrades.

### 4.1. ResNet-18

He et al. [11] proposed a residual network that applies residual concepts to the CNN model. ResNet showed that as the layer gets deeper, the gradient vanishing problem can be reduced using the residual block and hence bring performance gain. This architecture has shown superior performance in various image processing tasks than previous CNN models. In the CNN model, the receptive field of the unit in the deeper layer is larger than in the shallow layer. This is because as the layer deepens, the output unit is indirectly connected to a broader area of the input image [17]. However, as the layer deepens, structural problems such as gradient vanishing and over-fitting can easily occur. As shown on the right side of Figure 11, ResNet solves the aforementioned problems by connecting a detour path called identity shortcut connection between intermediate layers.

In the residual block, the input ‘x’ goes through the first convolution layer, the activation function (Relu) and the second convolution layer, outputting F(x). The output F(x) passes through the activation function after the addition with the initial input ‘x’. Due to this shortcut connection, even if F(x) has passed through the two layers and the parameter approaches 0 due to gradient vanishing, the added initial input ‘x’ remains and is transferred to the next layer. Therefore, even if the layer is deepened, the representation power does not fall short of the layer before the identity function and the performance is improved by training. In the ImageNet dataset, the ResNet model showed higher performance than the vanilla CNN model, and achieved better performance as the layer deepens. ResNet is a well-balanced CNN model widely used in recent computer vision tasks. As radar signals were transformed into a format of an image, we can train the detection problem on the dataset with these ResNet models. Through this, we can analyze the radar spectrogram data with the ResNet model and design an enhanced model. Compared to the optical image classification task, learning MDS and trajectory on a radar spectrogram is less complicated, so we use the ResNet-18 model with fewer layers.

First, we analyzed the performance of the model according to the information type of the spectrogram data. The radar signal is composed of complex numbers, containing the signal intensity and phase information, etc. In a related study, to utilize a deep learning-based optical image classification model, the signal value of the radar spectrogram was transformed into the color scale of the optical image. We assumed that the transformation without preserving radar data characteristics could distort or miss out on information and tested the performances according to the three different information form of the radar signals. The three types of radar information are represented as channel information: 1 channel of magnitude, 2 channels of real and imaginary and 2 channels of magnitude and phase. The magnitude is the square root of the sum of the squared real-value and the squared imaginary-value. The phase is obtained by taking the inverse tangent of the value obtained by dividing the imaginary value by the real value. The height and width sizes of the three spectrogram data were the same, and the accuracy is the average of five times measurements. Table 5 shows the classification accuracy according to the signal information form of radar spectrogram. The result shows the highest accuracy when the real and imaginary values of the radar signal are paired as two channels. Through this, it was confirmed that most features were maintained at the original form of the radar signal and that the change in signal form could lead to loss of information.

Next, we investigate the main features of the data that the model learns by adding two different noises to the dataset. One is the Gaussian noise that adds random value and the other is the uniform noise that adds the same value. The Gaussian noise image was created by generating a random variable following a standard normal distribution as the input size, multiplying it by a noise level representing noise intensity, and adding it to the original normalized image. The uniform noise image was created by setting the value of 1 to the input size, multiplying the noise level, and adding it to the original normalized image. Each value of the image does not exceed 1. Through Gaussian noise, we checked the model performance in the condition of where an arbitrary shape is added to the entire image, and in uniform noise, we checked the model performance when the sharpness of the MDS is reduced compared to the surrounding area. Each noise level was specified as a hyper-parameter by identifying the point at which the model’s performance begins to deteriorate significantly. Table 6 shows the performance of the model for two types of noises. The performance of the model decreases in both data as the noise level increases, but we see that the performance decreases sharply in the Gaussian noise, compared to the uniform noise. This result shows that low-level features in the radar spectrogram dataset are the most important features to the model when classifying UAVs.

The ResNet-18 model consists of a 5 convolution group and a second to fifth convolution groups consist of several basic blocks. The feature map size is halved after going through each convolution group.We analyzed the performance by changing the convolution group of the model and the basic block(layer)s within the groups. We checked the performance by sequentially removing the convolution groups from the output of the model. Table 7 shows the accuracy of the model for the final feature map size changed by the removal of the convolution group. The results showed the highest accuracy in the 8 × 8 feature map size with the 5th convolution group removed, and the performance continued to decline after that. This is the highest model performance in the optimal feature map size that reflects the characteristics of the data, and this feature is applied when designing a new model. We tested the performance by changing the number of basic blocks within the convolution group, but there was no significant trend. Through this experiment, we confirmed that the final feature map size is significant in learning the spectrogram data of the deep learning model, and that the additional depth of the layer does not significantly affect the performance improvement. Based on the above analysis, we design a lightweight model more suitable for real-time systems.

### 4.2. ResNet-SP

In the analyses with the ResNet-18 model, we checked that (1) the model mainly learns the low-level features of the radar spectrogram, (2) the size of the final feature map significantly affects the performance and (3) the depth of layers do not significantly affect the performance. Furthermore, we designed the ResNet-SP model for real-time systems by applying optimal settings based on these analyses and applying additional compression and stabilization methods.

Figure 12 is the architecture of ResNet-SP. We removed the 5th convolution group from the ResNet-18 model and kept the number of basic blocks the same.And we reduced the channels of each group by half. By applying a dilated kernel method, the computations were reduced because the smaller parameters are used in the model while keeping the receptive fields. To increase the learning stability of the model, we applied an outlier detection method that removes abnormal data in the learning process through the distribution of the data and the gradient clip method that reduces the influence of the anomalous data by limiting the norm of the gradient.

In a kernel, the receptive field signifies the area of the input image where the kernel attends to. The size of the receptive field is the same as the kernel size and the larger the size, the more the overall characteristics of the image can be obtained. However, if the kernel size is increased to obtain a wider receptive field, the number of parameters increases, which increases the computational time.

Dilated convolution is a method of adding zero-padding to the convolution kernel, which allows a wider receptive field while using the same number of parameters [18]. Figure 13 shows the receptive field images when 3 × 3, 5 × 5, and 3 × 3 kernels with dilation are applied. The 2-dilated 3 × 3 kernel has the same receptive field as the 5 × 5 kernel, with the same number of parameters as the 3 × 3 kernel. Through this method, the global feature can be extracted without increasing the number of parameters.

In the radar spectrogram, the MDS shape is formed around the main velocity by the main body, and the harmonic components appear parallel to around. Whereas the target is located locally in the visual image, the moving target signal of the radar spectrogram is time-varying, and the local characteristics of the MDS formed in specific areas of the entire image and the global characteristics of the harmonic component formed at various frequencies coexist. We tried to reduce the computational time without sacrificing the global feature. Therefore, we applied a dilated convolution kernel within the range that does not significantly degrade the performance. Table 8 shows the performance when the size of the 7 × 7 kernel of the model is reduced to a 3 × 3 kernel with the dilation. When dilation of 2 was applied to the 3 × 3 kernel, the performance was maintained. However, when dilations of 3 or more were applied, the performance was gradually decreased. This means that applying too large dilation will miss out the MDS information, the main feature that the model is supposed to learn. Therefore, at the end, we applied a dilation of 2 to the 3 × 3 kernel, instead of the 7 × 7 kernel.

Although many abnormal data were removed in the refinement step, abnormal data that interfere with learning still remained in our dataset. In a deep neural network using multiple layers, when such abnormal data comes in a batch, large weights from large loss are successively multiplied in the parameter update process, causing a gradient exploding problem. As a result, previous well-trained parameters change rapidly, which causes a dramatic accuracy drop. We applied a gradient norm clipping method [19] that constrains the maximum norm of gradients to reduce the impact of anomalous data interfering with the learning process. Figure 14 is the training loss curve before and after applying the gradient clipping method. In the figure, the red box represents the training loss value after 5000 iterations, and we can see that the variation has decreased after applying the gradient clipping. And the deviation of test accuracy was also reduced.

We additionally applied a method of removing abnormal data from the training process to increase the learning stability of the model. In our dataset, abnormal data are non-recorded or contaminated sections of the spectrogram. Most of these abnormal data were removed through the data refinement step, but some remain, interfering with the stable learning of the model. These abnormal data exist in every class. Anomaly detection is a research field that identifies outliers that deviate from the majority of normal data. There are various anomaly detection methods, but we applied the concept of a softmax model of end-to-end anomaly score learning [20]. This approach assumes that normal data appears at a relatively high frequency compared to anomalous data, and anomalous data appears at a lower frequency. When data of a specific class is input, the softmax value is much higher in that class than in other classes, and most of the normal data have this probability distribution. When normal data is entered into the model, the softmax value is significantly higher in one class. However, when abnormal data is entered into the model, the softmax value appears similar in several classes and the largest value is much smaller than the softmax value in normal data. Figure 15 is an example of the softmax distribution of normal and abnormal data. Using this characteristic, we excluded data in the training process if the highest softmax value of the input data does not exceed a certain threshold.

## 5. Experiment and Results

This section shows performances with the ResNet-18 and the ResNet-SP model on the radar spectrogram dataset. The performance measures we present are accuracy and computation time. The model accuracy is the average test accuracy of five runs and the standard deviation is also presented. The computational time is measured only during the training time, excluding the dataset generation procedure. We also presented inference time, which is the prediction time for one input.

As settings for the experiment, the model training was performed in Ubuntu with NVIDIA GeForce GTX Titan X edition GPU and a 3.6 GHz Intel Core i7-9700K CPU. We used the stochastic gradient descent (SGD) [21] with momentum [22] as an optimizer. The momentum coefficient was set to 0.9 which means 90% of the cumulated gradient from the previous step will be transmitted to the current step. The initial learning rate was 0.1 and the weight decay [23] coefficient for regularization was set to 1 × 10${}^{-4}$. We trained for 100 epochs in total, with a batch size of 64. We used the cross-entropy loss [24] for the final loss function.

Table 9 shows the computational time and accuracy of both models. As a result, ResNet-SP showed slightly higher accuracy with shorter computational time than ResNet-18, and the standard deviation of accuracy was also smaller, indicating that the learning stability was improved.

## 6. Conclusions

In this study, we recorded three different types of UAV signals and two different types of human activity signals in various scenarios using FMCW radar. Furthermore, we generated the radar spectrogram dataset with high diversity through STFT, the data refinement method, and the data augmentation method. Then, we analyzed the characteristics of the radar spectrogram dataset using the ResNet-18 and checked the optimal data form and model structure. In addition, we designed the ResNet-SP model, which is more suitable for real-time systems by compressing and stabilizing the ResNet-18 model. As a result of experimenting with both models with the same radar spectrogram dataset, the ResNet-SP showed higher stability, accuracy, and faster computational time than the ResNet-18 model. In future works, we hope to expand this study to a model that classifies UAV types by adding several additional UAVs and to improve the performance of the model using the acoustic spectrogram of the target along with the radar spectrogram. We also hope to improve the performance of the ResNet-SP model.

## Figures and Tables

**Figure 1 sensors-21-00210-f001:**
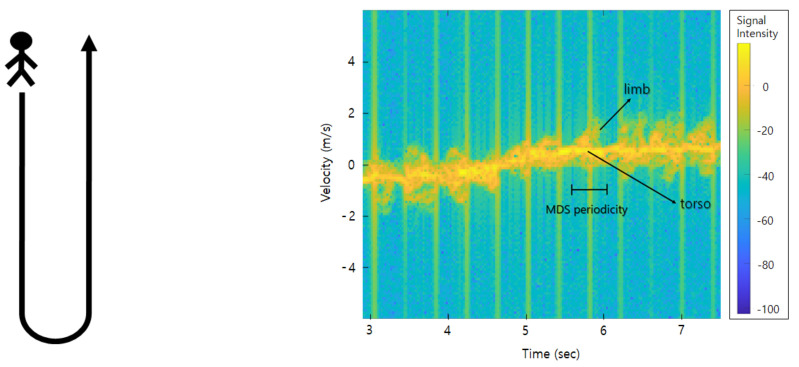
Spectrogram of a person walking: approaching from the front and turning away.

**Figure 2 sensors-21-00210-f002:**
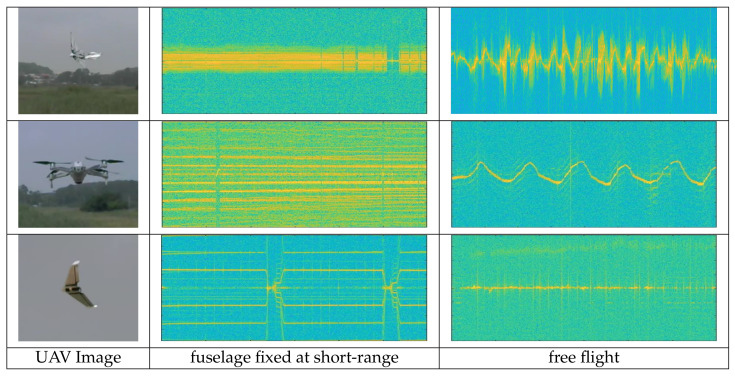
Spectrogram of Unmanned Aerial Vehicles (UAVs); wing-flap (**top**), quad-copter (**middle**), and fixed-wing (**bottom**).

**Figure 3 sensors-21-00210-f003:**
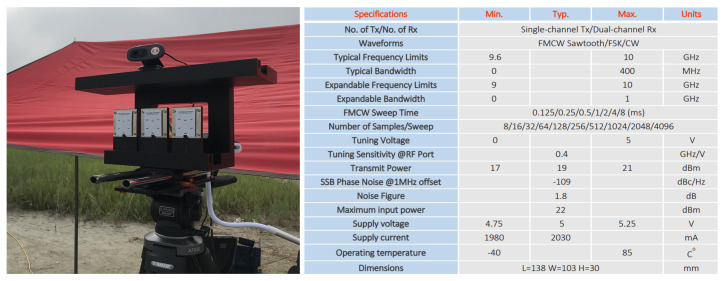
Staring mode X-band frequency modulated continuous wave (FMCW) radar (Ancortek’s SDR-KIT 980AD2) and Specification.

**Figure 4 sensors-21-00210-f004:**
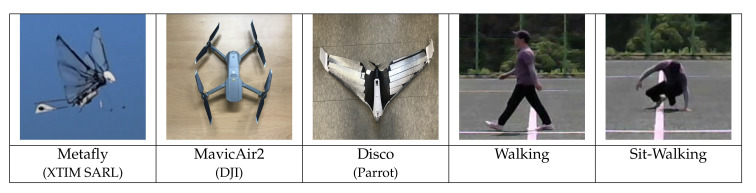
Five target Images; three types of UAVs and two different human activities.

**Figure 5 sensors-21-00210-f005:**
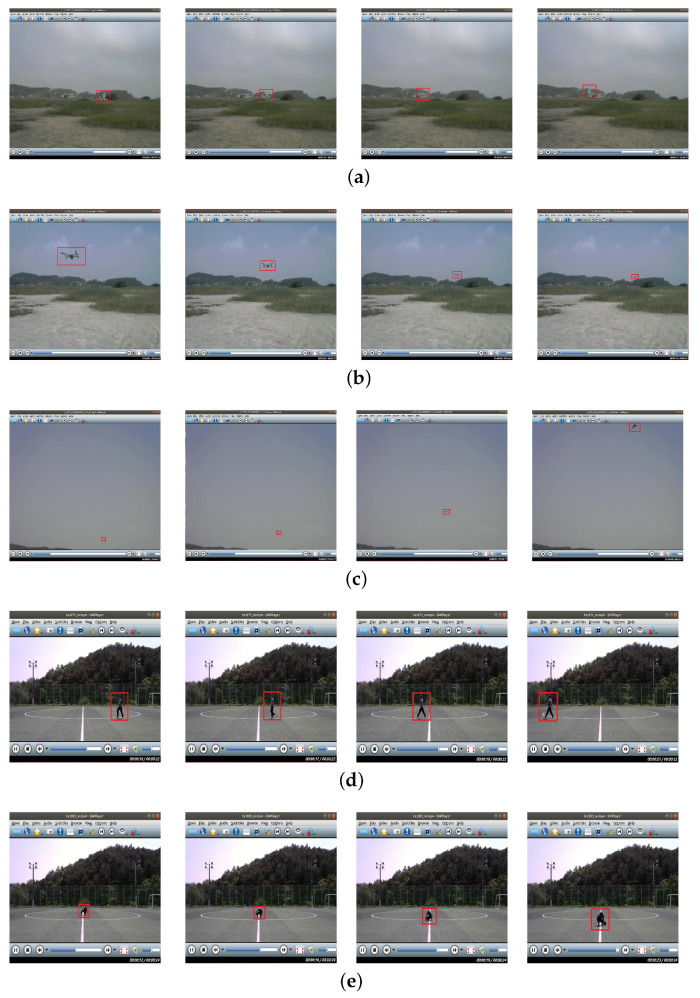
Sequential video frames of a specific movement for each target: (**a**) Metafly flight from right to left; (**b**) Mavic Air 2 flight from front to back; (**c**) Disco flight back to front; (**d**) Walking right to left; (**e**) Sit-Walking back to front.

**Figure 6 sensors-21-00210-f006:**
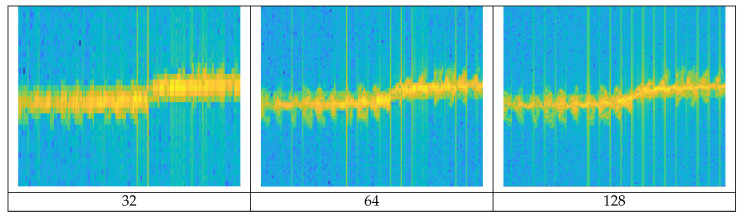
Spectrogram resolution of walking according to window size; as the window size increases, the frequency resolution increases.

**Figure 7 sensors-21-00210-f007:**
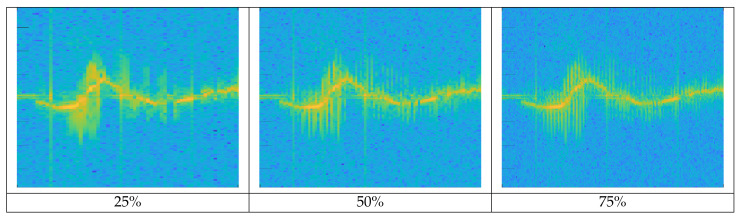
Spectrogram shape of Metafly flight according to window overlap ratio: approx. 24 wing beats per second of Metafly appear more clearly as the window overlap ratio increases.

**Figure 8 sensors-21-00210-f008:**
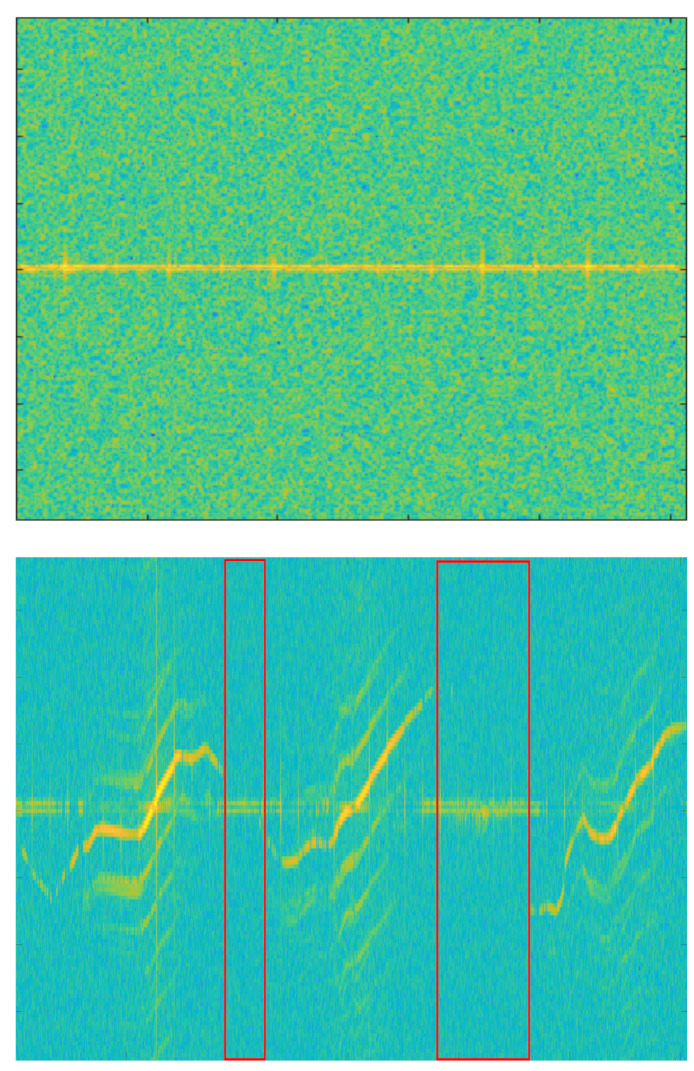
Spectrogram of background clutter (**top**) and Mavic Air 2 (**bottom**). The red box is the non-recorded section of the target.

**Figure 9 sensors-21-00210-f009:**
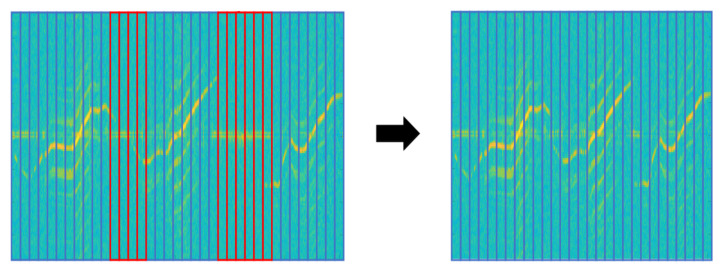
Spectrograms of Mavic Air 2 before refinement (**left**) and after refinement (**right**). red box: chopped image with an average intensity below threshold, blue box: chopped image with an average intensity above threshold.

**Figure 10 sensors-21-00210-f010:**
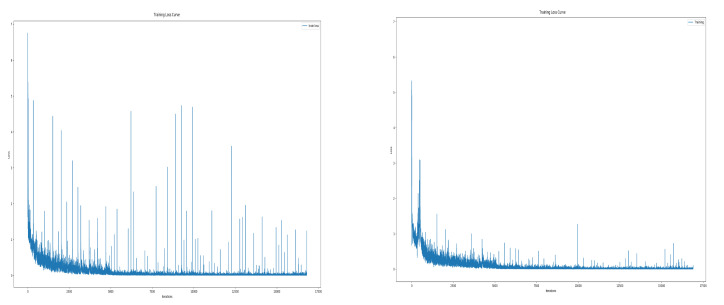
Training loss curve; before data refinement (**left**), after data refinement (**right**).

**Figure 11 sensors-21-00210-f011:**
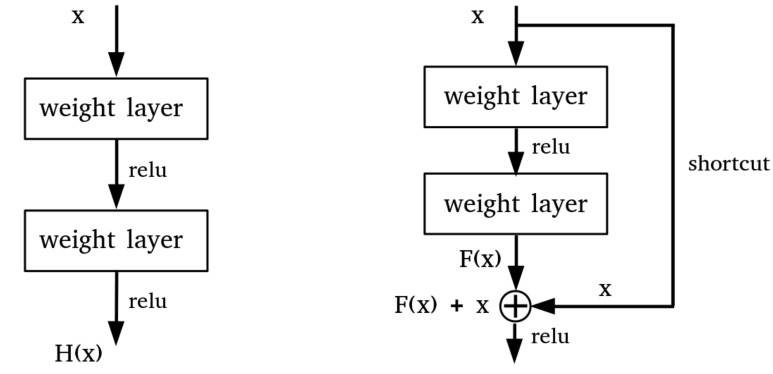
Plane CNN layers (**left**), Residual block (**right**).

**Figure 12 sensors-21-00210-f012:**
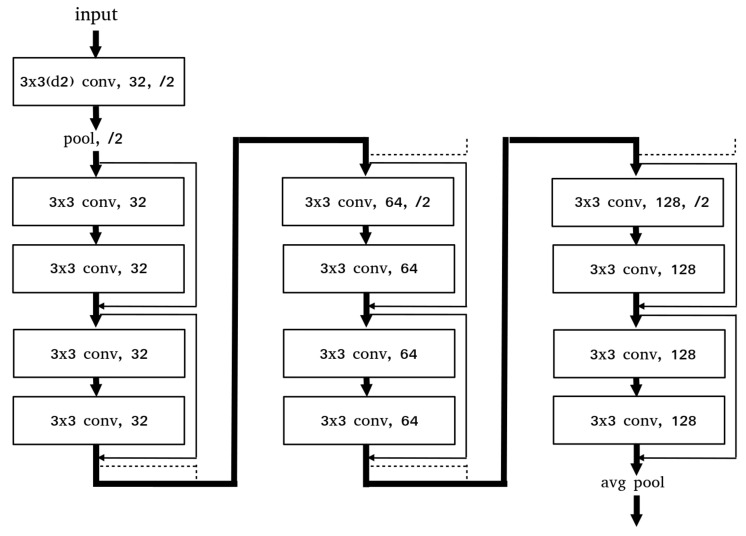
ResNet-SP Architecture.

**Figure 13 sensors-21-00210-f013:**
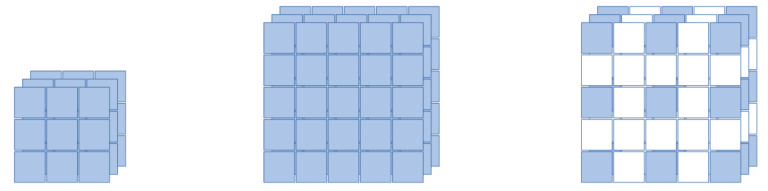
The receptive field images: 3 × 3 kernel (**left**), 5 × 5 kernel (**center**), 2-dilated 3 × 3 kernel (**right**).

**Figure 14 sensors-21-00210-f014:**
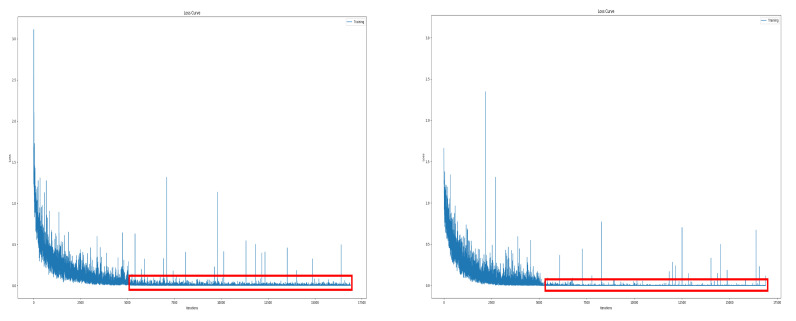
The training loss curve before (**left**) and after (**right**) applying the gradient clipping method.

**Figure 15 sensors-21-00210-f015:**
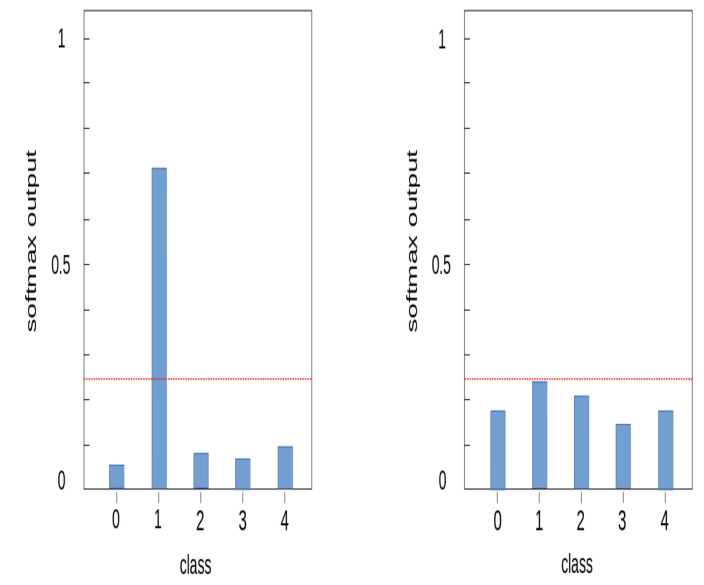
Softmax output distribution on data; normal data (**left**), outlier data (**right**).

**Table 1 sensors-21-00210-t001:** The movements for each target and the settings for recording. L is left, R is right, B is back, F is forth and C is concentric.

Parameter	Metafly	Mavic Air 2	Disco	Walking	Sit-Walking
Alt./Range (m)	0–10/0–10	0–10/0–100	10/0–100	0/0–100	0/0–100
Movement	Free flight	Free flight	Circular-flight (L ↔ R, B ↔ F, C)	Free	Free

**Table 2 sensors-21-00210-t002:** The spectrogram width size of 1 recorded radar signal.

Division	Mavic Air 2	Disco
Before refinement	995	339
After refinement	747	168
Removal percentage	25%	50%

**Table 3 sensors-21-00210-t003:** Data augmentation method applied when generating training data.

Category	Window Size	Window Overlap	Vertical Flip	Total
(Specification)	(128, 256, 512)	(50%, 70%, 85%)	(O, X)	
Original signal	×3	×3	×2	×18

**Table 4 sensors-21-00210-t004:** The number of data for each class in the radar spectrogram dataset for five low altitude, slow speed, and small radar cross-section (LSS) targets.

Class	Metafly	Mavic Air 2	Disco	Walking	Sit-Walking	Total
Train	2142	2176	2196	2136	2112	10,762
Test	219	218	206	198	195	1096

**Table 5 sensors-21-00210-t005:** Classification accuracy of ResNet-18 model according to the signal form of radar spectrogram.

Channels	Accuracy (%)
1 (Magnitude)	75.98
2 (Real, Imaginary)	79.88
2 (Magnitude, Phase)	54.53

**Table 6 sensors-21-00210-t006:** Accuracy of ResNet-18 on the spectrogram dataset with two different noises.

Noise Level	Accuracy on Gaussian Noise (%)	Accuracy on Uniform Noise (%)
0.01	76.20	80.80
0.03	66.30	80.90
0.05	40.25	75.71

**Table 7 sensors-21-00210-t007:** Accuracy by number of convolution groups and layers in ResNet-18.

Conv. Groups	Numbers of Layers	Feature-Map Size	Accuracy (%)
5	18	4 × 4	79.88
4	14	8 × 8	81.43
3	10	16 × 16	75.38

**Table 8 sensors-21-00210-t008:** The accuracy by the kernel shape in the first convolution group on model.

Kernel size	7	3	3
Dilation	1	2	3
Receptive field	7	5	7
Accuracy (%)	79.88	80.26	79.33

**Table 9 sensors-21-00210-t009:** Mean accuracy and standard deviation of five measurements, and computational time.

Models	Inference Time (ms)	Training Time (s)	Accuracy (%)	Standard Deviation
ResNet-18	2.68	640.39	79.88	0.0204
ResNet-SP	1.98	242.22	83.39	0.0115

## Data Availability

Restrictions apply to the availability of these data. Data was obtained from Hanwha Systems Co. and are available from the authors with the permission of Hanwha Systems Co.

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
