# Peer review of "Radar-Spectrogram-Based UAV Classification Using Convolutional Neural Networks"

_sensors, 2020, doi:10.3390/s21010210_

Round 1

Reviewer 1 Report

Summary:

This paper describes a detection method of UAVs from deep learning approach. The authors proposed a CNN-based categorization method applying radar spectograms. They designed a network using ResNet and evaluate the accuracy of the detection method. 

Comments:

  • There are many grammatical errors in the manuscript. Please correct those.
  • It is hard to understand the overall research design from the manuscript. The authors need to reveal the originality and should make a straight story of the explanation of the main idea. Especially, it is not clear that what the main contribution of the research is and what the main difference from the related researches. Authors need to revise the structure of the paper.
  • It is hard to understand the overall flow of the proposed method.
  • Why do the authors need to gather the data of walking persons? It is hard to understand the requirement.
  • The manuscript does not include any references to figures and tables. Those should be referenced in sentences of the manuscript with detailed explanation of the figures/tables.
  • What is the main originality of the proposed method?
  • It is hard to understand the accuracy in the evaluation. How do the authors evaluate the accuracy?

Reviewer 2 Report

The manuscript aim to contribute to the solution of the problem of drone classification in the radar micro-Doppler images.

The manuscript is in general well written with only few minor typos (capitalization and punctuation related) and reasonable structure.

The main concern about the manuscript in the reviewer's opinion is related to lack of clarity in novel contribution formulation. As per presented results reader will struggle to see what this work proposes towards addressing the known challenges in drone classification, correctly pointed by authors in the introductory sections. 

From the technical perspective the comments are as follows:

Authors need to take more efforts in addressing leakage artefacts in spectrograms. Due to these artifacts applied ML technique may have more issues than expected with convergence and may not be able to achieve good performance (83% is quite far from current over 90% level of performance in classification tasks). 

It is doubtful (and has to be discussed more) whether augmentation using different overlaps,  resolutions and flipping is sufficient to avoid overfitting. Strategy of resizing has to be discussed too, as it can neglect augmentation efforts using resolution change.

Performance of the proposed modified ResNet is not analysed and discussed enough to understand reasons for reduced performance. Results with noise need thorough analysis too; in particular why performance drops so low and why more for gaussian noise.

Selection and practical meaning of decided level noise addition is not clearly explained. Noise levels have to be presented in more practicals terms rather then generic "noise level" 0.01 or 0.05.

The manuscript also does not provide critical comparison with other known (published) results in this area.

Round 2

Reviewer 1 Report

Authors encouraged to revise the manuscript. All questions and comments are responded by the authors.

What is concerned is English text editing. Please carefully revise grammatical error or typos again such as found in the line 69 "models that learns MDS that is ...".

Reviewer 2 Report

The manuscript has been improved in this round; however, the major concerns have not been fully addressed. To reiterate my previous comments:

Novelty - (1) UAV dataset claimed as novel is not publicly available, details of the diversity are not provided. How this is addressing existing challenges? (2) Analysis of characteristics of radar spectrograms is of doubtful value for readers. What is novel in this analysis? (3) Design of the ResNet-SP is a novel contribution when it is shown that it is addressing some known problems - this is not shown in the manuscript. Please add more clear statement of novelty to the manuscript.

Comment about artefacts has not been responded. This is one of major issues which authors are trying to address artificially through post-processing and the model tuning while it should be done during MDS calculation.

Please add discussion about augmentation to the text.

Comment about discussion of the reduced performance is not addressed as well as the comment about noise parameters. It is still not clear why the commonly used Gaussian noise brings performance to a very low level, making model effectively not suitable for practical applications. Noise values in an absolute representation are not helpful for readers too, please provide equivalent SNRs for wider comparison between different techniques and data.

Comment about comparison is not addressed. Please provide comparison either at the data level (i.e. using known data to quantify your technique performance) or apply another known technique to your data to characterise improvements that you aim to introduce by developing your novel approach.
